


# Distance Scaling Method for Accurate Prediction of Slowly Varying Magnetic Fields in Satellite Missions

P. P. Zacharias[1], E. G. Chatzineofytou[1], S. T. Spantideas[1] and C. N. Capsalis[1]

[1] National Technical University of Athens, 15780, Greece

*Correspondence to*: P. P. Zacharias (panoszach93@gmail.com)

**Abstract.** In the present work, the determination of the magnetic behavior of localized magnetic sources from near field measurements is examined. The distance power law of the magnetic field fall-off is used in various cases to accurately predict the magnetic signature of an EUT consisting of multiple AC magnetic sources. Therefore, parameters concerning the location of the observation points (magnetometers) are studied towards this scope. The results clearly show that these parameters are independent of the EUT's size and layout. Additionally, the techniques developed in the present study enable the placing of the magnetometers close to the EUT, thus achieving high SNR. Finally, the proposed method is verified by real measurements, using a mobile phone as an EUT.

## I. Introduction

The prediction of the magnetic behavior of localized magnetic sources plays a significant role in many scientific and engineering applications and has been widely discussed [1-6]. In particular, "magnetic cleanliness" in space missions is a scientific field that requires the determination of the magnetic signature of all sources included in the equipment of a spacecraft. Several recent and upcoming spacecrafts' missions (Solar Orbiter, Juice) are destined to perform in situ AC magnetic field measurements of planetary objects [10], as well as measure dynamic effects on test masses (Lisa Pathfinder) [11]. The measuring equipment (magnetometers) must be placed onboard at "magnetically clean" specification points, namely where the magnetic field contributions of the spacecraft is kept below $0.1 - 1$ nT [1]. In order to determine these "magnetically clean" specification points, strict ground verification requirements must be satisfied. For these purposes, several methods have been developed, involving measurements and modeling procedures in both unit and system level [4]. These techniques involve the study of the magnetic sources that have the same magnetic signature as the equipment mounted on the spacecraft.

In order to accurately predict the magnetic behavior of any Equipment under Test (EUT), near magnetic field measurements are performed. The case of Direct Current (DC) magnetic sources has been thoroughly studied and many approaches have been proposed. Specifically, the dominant approach involves the Magnetic Dipole Modeling (MDM) [7-9]. Conventionally, the magnetic sources associated with the spacecraft's equipment can be considered as magnetic dipoles, thus assuming that their magnetic field's distance power law approximates the $r^{-3}$ law [15]. However, this assumption is not always valid since several parameters of the magnetic sources may have significant impact on the EUT's magnetic signature. Therefore, these methods cannot be used in cases where the EUT consists of numerous magnetic sources. Furthermore,



upcoming space missions have additional demanding requirements regarding AC magnetic cleanliness, focusing on the range DC – 1 MHz, where the magnetic field can be treated as quasi-static [12].

Previous techniques aim at the prediction of the worst case scenario, i.e. the maximum possible value of the magnetic field at the specification points. This entails the prevention of the magnetic field's underestimation, whereas the overestimation of the field is basically considered less significant. Nevertheless, in many cases the accurate prediction of the magnetic field is required, regardless of under/over estimation.

The present work focuses on the development of methods and techniques that allow the accurate determination of the magnetic behavior of an EUT, which consists of several magnetic sources (current loops). Low frequency (up to 1 MHz) magnetic sources have been employed in several cases to compose an EUT. The magnetic field is calculated using the Biot - Savart law and the magnetic vector potential theory. At this range of frequencies, these two physical laws can be used interchangeably for calculation of the near magnetic field [16]. Eventually, this study will allow the accurate extrapolation of the EUT's magnetic field at the intended specification points.

The magnetic field's power law with respect to the observation distance is examined. Without loss of generality, two observation points (dual magnetometer technique) are adequate in order to model the power law of the magnetic field [14]. Thereafter, the fall-off of the magnetic field is employed in order to accurately predict the magnetic behavior of an EUT at various distances. The scope of this work is to study the impact of the magnetometer's locations on the prediction of the magnetic field at various distances.

Typically, the magnetic field measurements are conducted in close proximity to the EUT for high Signal to Noise (SNR) purposes. The magnetic behavior of the EUT is assumed to follow the dipole's fall-off when the observation point is located more than about 5 times the maximum loop's dimension [4]. Below this threshold, where the magnetometers are placed, the power law of the magnetic field may deviate from the aforementioned law.

The rest of the paper is organized as follows: in Section II, the mathematical background is briefly presented and the mathematical problem under consideration is formulated. In Section III, several simulation results concerning the magnetic behavior of an EUT are carried out. Moreover, real measurements are used in order to validate the simulation results.

## II.      Modeling Procedure

Any equipment that may influence the magnetic cleanliness requirements at the specification points can be modeled as a "black box" with current sources inside [13]. The non-ideal EUT consists of numerous current sources, whose positions inside the box are unknown in advance. For the purposes of the present study, the EUT is assumed to be centered at the origin with dimensions $b \times b \times b \ m^3$. The position $(x_j, y_j, z_j)$ of the current source $j = 1, 2, \dots N$ inside the box is randomly selected. In Fig. 1, an EUT containing several magnetic sources is depicted. Each magnetic source is assumed to be a square current loop of area $A = a^2$ (side $a$) and current $I$. The centers of the magnetic sources are randomly scattered following a


uniform distribution in the interval $[(-b+a)/2, (b-a)/2]$ $m$ in each direction $(x, y, z)$, thus assuring that the loops' wires are located inside the box.

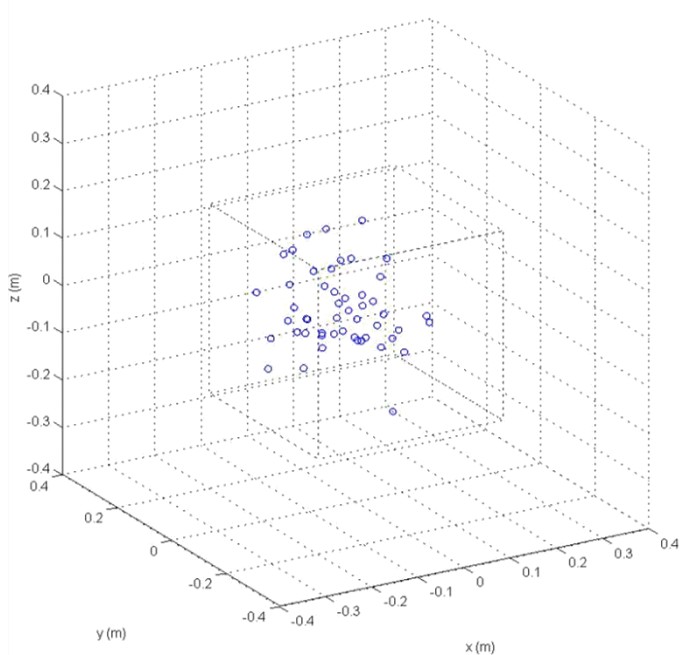

**Fig.1.** EUT consisting of multiple magnetic sources

5      The parameters of the magnetic sources are tabulated in Table 1. All magnetic loops are aligned at the $z$-axis, formulating the worst case scenario in terms of magnetic field strength. Finally, several simulations concerning the random positioning of the sources inside the box have been carried out. Evidently, the location of the sources inside the box has a significant impact on the magnetic behavior of the EUT, leading to a substantial deviation from the $r^{-3}$ distance power law.

TABLE I

PARAMETERS OF THE MAGNETIC SOURCES

| **Parameter** | |
|---|---|
| Loop area | $\pi \ cm^2$ |
| Loop current | $1 \ mA$ |
| Loop position | Random |
| Loop orientation | On $z$-axis |
| Number of loops | 50 |
| Box width | $0.40 \ m$ |
| Number of independent trials | 12 |



As already mentioned, the analysis will be based on two observation points, where the magnetometers are placed. The observation points are assumed to be on the *z*-axis, with varying distances from the center of the box, as shown in Fig. 2.

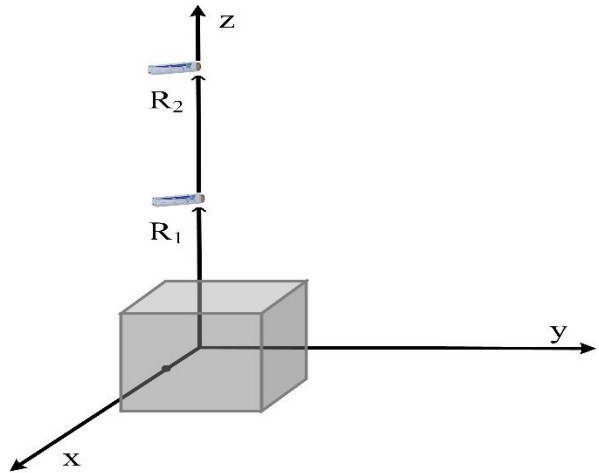

**Fig.2.** Observation points (magnetometers) in close proximity to the EUT

In order to calculate the magnetic field at the observation points, following the Finite Element Model (FEM) method, the magnetic sources are divided into *W* finite current elements of length *dl* that carry a current *I* (each current loop is divided in 200 elements as a safety margin). Then, the magnetic field of each element, $w_i$, *i=1…W* is calculated via the

Biot - Savart law:

$$d\vec{B}_i = \frac{\mu_0}{4\pi} \cdot \frac{\left(I \cdot d\vec{l}_i \times \vec{r}_i\right)}{|\vec{r}_i|^3} \tag{1}$$

where $\vec{r}_i = (\vec{x} - \vec{x}'_i)$ is the distance vector between the observation point and the current element $w_i$. Then, the magnetic field of all sources is calculated as the superposition of the magnetic field of all current elements, that is:

$$\vec{B} = \sum_{j=1}^{N} \sum_{i=1}^{W} d\vec{B}_{ij} \tag{2}$$

Alternatively, the magnetic field can be calculated employing the magnetic vector potential:

$$\vec{A}(\vec{r}) = \frac{\mu_0}{4\pi} \cdot \int_V \vec{J}(\vec{x}') \cdot \frac{e^{-jkr}}{r} dV' \tag{3}$$




where $k = \omega/c$ is the wavenumber and sinusoidal time dependence of the current density is assumed $(\vec{J}(\vec{x}, t) = \vec{J}(\vec{x}) \cdot e^{j\omega t})$. The magnetic field is then calculated via

$$\vec{B} = \nabla \times \vec{A} \qquad (4)$$

For low frequencies ($k \to 0$), the vector potential theory and the Biot – Savart law can be used interchangeably.

Once the EUT has been initialized, the magnetic field is calculated at specific observation distances using either Biot-Savart law or magnetic vector potential. Subsequently, the values of the magnetic field at the observation points are employed in order to calculate the power law. For the purposes of this study, exponential interpolation is used in order to 
10 determine the power law of the magnetic field fall-off $r^{-n}$. At least two observation points are needed in order to apply the exponential interpolation method. The magnetic field is considered to have the following form:

$$B = |\vec{B}| = \frac{a}{r^n} \qquad (5)$$

The above equation can be written:

$$\ln B = \ln a - n \cdot \ln r \qquad (6)$$

Hence, the magnetic field values at two observation points are adequate to determine the coefficients $a$ and $n$. The observation distance from the center of the EUT, as well as the spacing between the two magnetometers, are varied. It is worth mentioning that this analysis can be carried out for each component of the magnetic field ($B_x$, $B_y$, $B_z$) individually. 
In each case, the power law is used to extrapolate the magnetic field at the specification points of interest, where the sensitive measure equipment is located. The relative error ($R_E$) between predicted and theoretical field values (calculated from Eq. (2) or Eq. (4)) is examined and is used as a criterion to assess the overall prediction process:

$$R_E = \frac{|B_{predicted} - B_{theoretical}|}{|B_{theoretical}|} \cdot 100\% \qquad (7)$$

### III.     Simulation Results

In the simulation results presented in the present section, the EUT is considered to be a box with dimensions $40 \times 40 \times 40 cm^3$. The impact of the observation distance from the center of the EUT on the accuracy of the extrapolated magnetic field is examined. Moreover, the spacing between the two magnetometers is varied, parameter that further affects the precision of




the extrapolation. The value of the magnetic field is predicted at several distances from the center of the EUT. As already mentioned, the prediction method is assessed by calculating the relative error between the predicted and theoretical field values. This procedure is carried out for 12 individual cases of the box's layout. Subsequently, a study of the impact of the box's size on the accuracy of the prediction is carried out. Finally, a real EUT is examined in order to validate the above

simulation results.

### A.   Extrapolation at larger distances using exponential interpolation

The aforementioned modeling procedure is used to accurately predict the value of the magnetic field at specification points far away from the center of the EUT. The extrapolation distance is chosen to be $3m$ (7.5 times the size of the box), while magnetometers' positions and spacing between them are the modified parameters. Firstly, keeping the spacing

between the magnetometers fixed at $30cm$, their positions are varied. For each case, the theoretical values of the magnetic field at the observation points are obtained and employed in order to model the magnetic field's fall-off, by estimating the coefficients $a$ and $n$. In Fig. 3, the relative error between the predicted and theoretical magnetic fields at the extrapolation distance of $3m$ with respect to the closest magnetometer's distance from the center of the EUT is depicted. The 12 curves correspond to the 12 individual cases of the box's layout.

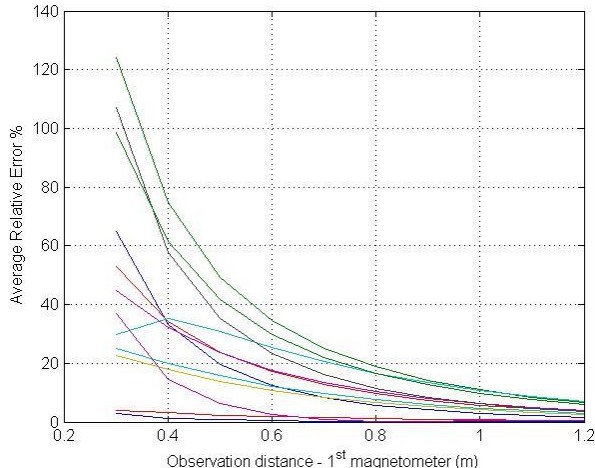

**Fig.3.** Magnetic field's relative error with respect to the first magnetometer's position with fixed spacing $30cm$ between the magnetometers – 12 independent trials.

As readily observed from Fig.3, the loops' positions inside the box play a significant role in the extrapolation

procedure. In some cases the loops are concentrated around the center of the box, thus achieving the dipole's $r^{-3}$ law in close proximity to the EUT. On the contrary, in other cases the loops are asymmetrically located close to the box's edges, leading to over or under estimation of the magnetic field at the extrapolation point. From Fig.3 it is evident that when the first magnetometer is placed at larger distances from the center of the EUT, the relative error is reduced.



Then, the influence of the spacing between the magnetometers on the accuracy of the extrapolation method is studied. For this purpose, the above procedure is carried out for $40cm$, $50cm$ and $60cm$ spacing between the magnetometers. The average relative error of 12 individual cases for varying spacing between the magnetometers is depicted in Fig.4. The red line corresponds to the relative error of 10%, which is considered an adequate threshold for the accuracy of the prediction. It is evident that when the first magnetometer is located at approximately 2 times the maximum dimension of the box, the relative error is below the acceptable value, despite the fact that the power law has not yet converged to $r^{-3}$.

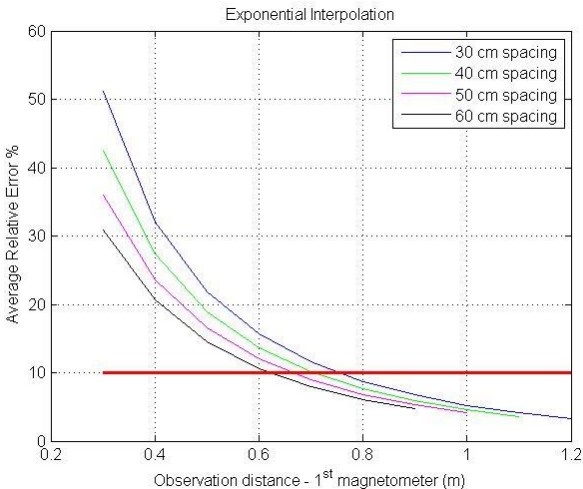

**Fig.4.** Magnetic field's average relative error with respect to the first magnetometer's position with fixed spacing $30cm$, $40cm$, $50cm$ and $60cm$ between the magnetometers.

Apparently, when the spacing between the magnetometers is increased, the relative error in the prediction of the magnetic field decreases regardless of the position of the magnetometers, designating that the first magnetometer can be placed closer to the EUT. In particular, when $60cm$ spacing is used, the first magnetometer can be placed at approximately 1.5 times the maximum dimension of the box, achieving higher SNR values.

### B. Extrapolation at larger distances using smoothing technique

In this subsection, a smoothing technique is employed in order to further increase the accuracy of the prediction of the magnetic field at $3m$ away from the center of the EUT. At the extrapolation distance of $3m$, the magnetic field fall-off approximates the dipolar scaling power, following the $r^{-3}$ distance power law [4, 15]. Therefore, the factor 3 can be used to smooth the estimated values of the coefficient $n$ as follows:

$$B = \left|\vec{B}\right| = \frac{a}{r^{(n+3)/2}} \qquad (8)$$





The above equation signifies that the magnetic field fall-off will exhibit the $r^{-3}$ distance power law dependence at smaller distances, thus enabling the placing of the magnetometers even closer to the EUT.

The magnetic field's relative error with respect to the closest magnetometer's distance from the center of the EUT, with fixed spacing at $30cm$, $40cm$, $50cm$ and $60cm$ using the aforementioned smoothing technique is depicted in Fig.5.

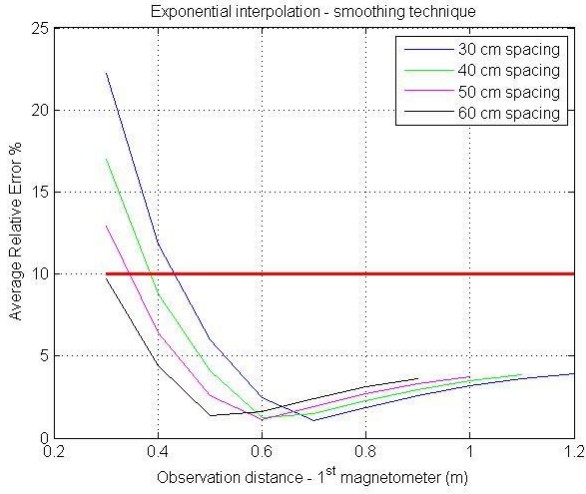

**Fig.5.** Magnetic field's average relative error with respect to the first magnetometer's position with fixed spacing $30cm$, $40cm$, $50cm$ and $60cm$ between the magnetometers, using the smoothing technique.

As readily observed, the smoothing technique significantly diminishes the relative error for all magnetometers' positions. The magnetometers can be located even closer to the EUT than before, achieving higher SNR values. The first magnetometer can be placed at approximately 1 times the maximum dimension of the box. As a result, this smoothing technique is suitable when the observation points are located relatively close to the EUT and the prediction of the magnetic field at larger distances is attempted. As the observation points are placed further away from the EUT, this method becomes less efficient and converges to the former one.

### C. Extrapolation between the magnetometers

Similar analysis can be carried out when the extrapolation point is located between the two magnetometers. Specifically, the accurate prediction of the value of the magnetic field at $1,1m$ away from the center of the EUT, using both exponential interpolation and smoothing technique is implemented. The verification point is located between the two magnetometers, corresponding to an intermediate prediction of the magnetic field's behavior. During this procedure, the spacing between the magnetometers is fixed at $60cm$ and their positions are varied. The magnetic field's relative error with respect to the closest magnetometer's distance from the center of the EUT, is depicted in Fig.6.





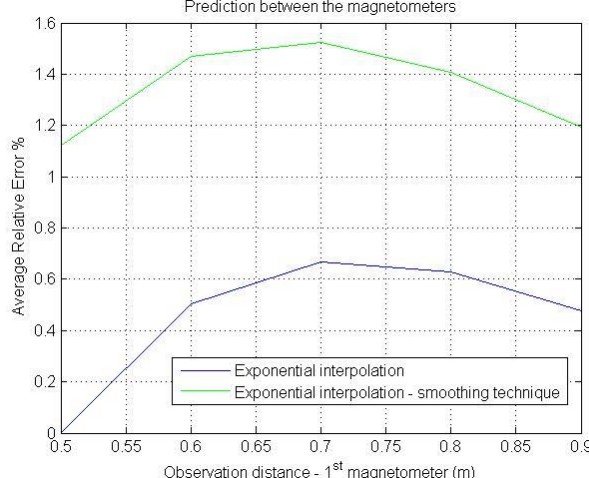

**Fig.6.** Magnetic field's average relative error with respect to the first magnetometer's position with fixed spacing 60cm between the magnetometers.

Evidently, the relative error is sufficiently small regardless of the position of the magnetometers, allowing accurate

prediction of the magnetic field at the intermediate specification points. Moreover, the smoothing technique induces higher relative error. This is due to the fact that at this extrapolation point, the magnetic field's power law severely deviates from the $r^{-3}$ law. As a result, there wouldn't be further improvement on the prediction's accuracy compared to the exponential interpolation technique.

### D.    Comparison of magnetic field's prediction methods

Previous extrapolation techniques employ either $r^{-3}$ or $r^{-2}$ distance power laws, leading to underestimation or overestimation of the magnetic field, respectively. A more advanced method focuses on the prevention of the magnetic field's underestimation, taking into account measurements from a single observation (verification) point. Specifically, the modeling involves a transition of the magnetic field distance power law from $r^{-2}$ to $r^{-3}$ after a predefined break distance [13]. In order to compare the aforementioned method to the techniques presented in this paper, the prediction process of the

magnetic field with respect to the distance was implemented at several scenarios. In Fig. 7, an indicative result concerning a specific box layout is shown. The first magnetometer is placed at $50cm$, namely at approximately 1 times the maximum dimension of the box, whereas the spacing between the magnetometers is fixed at $60cm$, namely at approximately 1.5 times the maximum dimension of the box. Additionally, the magnetometer at $50cm$ operates as a verification point and the break distance is chosen to be $1m$ (2.5 times the dimension of the box).

As is evident from Fig. 7, the transition from $r^{-2}$ to $r^{-3}$ technique overestimates the magnetic field up to the break point, while underestimating it between the break and the extrapolation distance of $3m$. However, the exponential interpolation and the smoothing techniques achieve a more accurate prediction at the complete distance range.



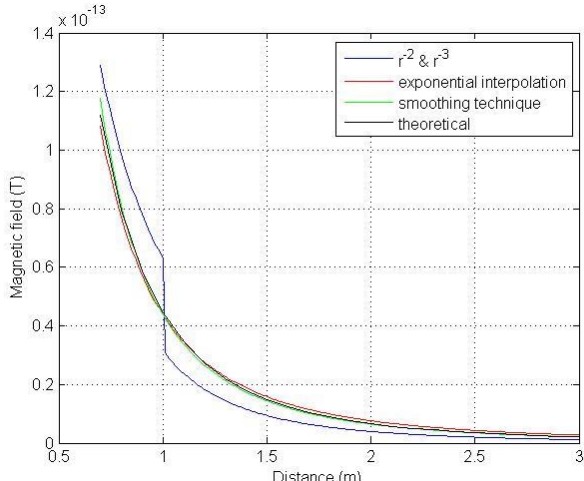

**Fig.7**. Magnetic field's prediction with respect to the distance with fixed magnetometers' locations at $50cm$ and $110cm$, using several techniques.

### E. Different box's dimensions

In order to study the impact of the EUT's size on the prediction method, the above comparison is carried out for different box's dimensions. The magnetometers' positions and the spacing between them are selected according to the aforementioned analysis. In the first case the box's dimensions are $20 \times 20 \times 20cm^3$ while in the second case the box's dimensions are $60 \times 60 \times 60cm^3$.

The prediction of the magnetic field with respect to the observation distance, using several techniques, is depicted in Fig.8 and Fig.9 for the different EUT's dimensions. In each case, the first observation point is located at approximately 1 times the dimension of the box and the spacing between the two magnetometers is approximately 1.5 times the maximum dimension of the box.

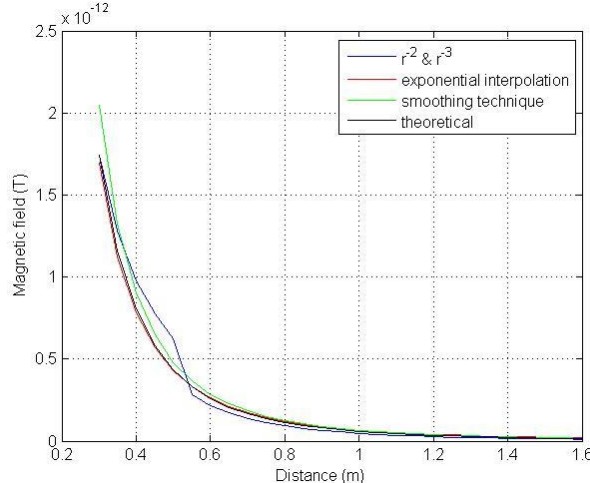

**Fig.8.** Magnetic field's prediction with respect to the distance with fixed magnetometers' location at $25cm$ and $55cm$, using several techniques (dimensions $20x20x20cm^3$).


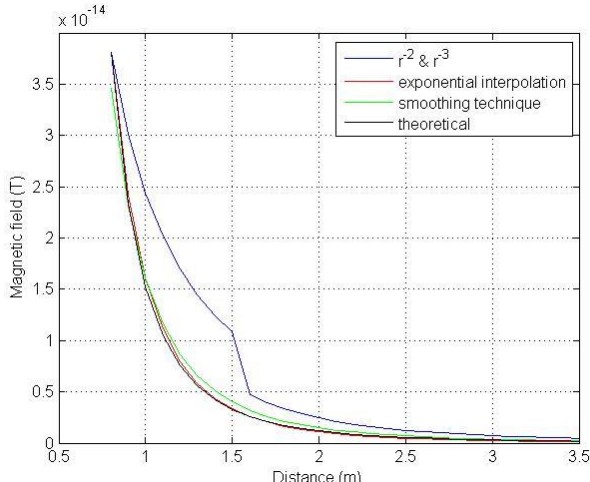

**Fig9.** Magnetic field's prediction with respect to the distance with fixed magnetometers' location at 80cm and 160cm, using several techniques (dimensions 60x60x60$cm^3$).

Apparently, the transition from $r^{-2}$ to $r^{-3}$ technique fails to provide an accurate prediction of the magnetic field at intermediate distances in both cases, even if the prediction at larger distances induces a small relative error. Nevertheless, the two proposed methods achieve a low relative error at all distances.

## F.   Verification using real measurements

For the purposes of verification of the proposed techniques, a real EUT is examined. In particular, near magnetic field measurements of a mobile phone with dimensions approximately $15 \times 8 \times 1 cm^3$ have been obtained. The measuring setup is shown in Fig. 10. It is worth mentioning that only one magnetometer is used to measure the near magnetic field values of the EUT.

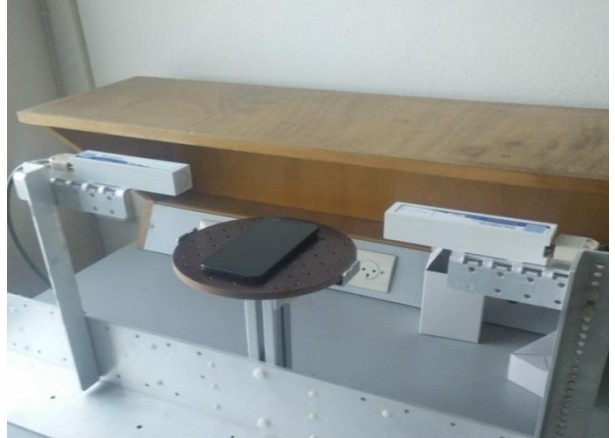

**Fig10.** Measuring setup and mobile phone used as EUT.





The prediction of the magnetic field employing the proposed techniques is implemented according to the above-mentioned analysis and the results are shown in Fig. 11. Evidently, the results confirm the simulated scenarios presented in the previous subsections.

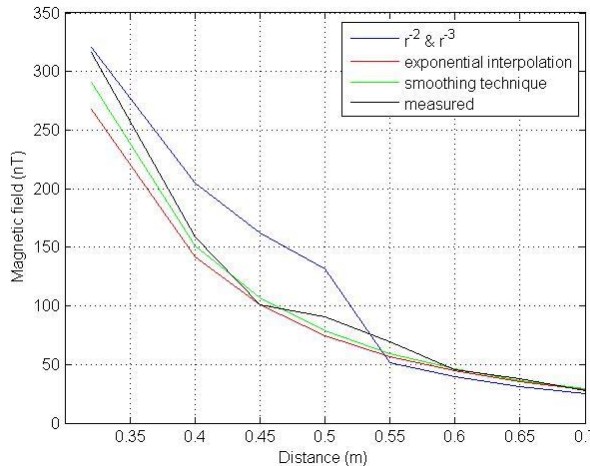

**Fig11.** Magnetic field's prediction with respect to the distance with fixed magnetometers' location at $26\,cm$ and $45\,cm$, using several techniques (mobile phone used as EUT).

## IV.    Conclusions

In reality, an EUT consists of numerous magnetic sources, whose positions are not necessarily known in advance. In the present work, two novel techniques have been presented in order to predict the magnetic behavior of an EUT consisting of multiple AC current loops. The field's fall-off is estimated by measuring the near magnetic field at two observation points (magnetometers). The modeling of the power law is obtained by exponential interpolation employing these values. Furthermore, a smoothing technique is proposed in order to further increase the accuracy of the prediction. Both techniques allow the determination of the magnetic field at several distances (extrapolation points).

The relative error between predicted and theoretical field values serves as an adequate criterion in the overall assessment of the prediction process. The simulation results clearly show that the observation distance, along with the spacing between the magnetometers play a significant role in the relative error of the prediction. Consequently, the aforementioned parameters (in addition to the proposed techniques) can be carefully chosen in order to diminish the relative error in the prediction. Additionally, the magnetometers can be placed in close proximity to the EUT, thus achieving higher SNR. It is evident that these parameters are independent of the EUT's size (relative to the box's dimensions) and of its layout.

Future analysis can be carried out regarding several parameters that may further influence the magnetic signature of real EUTs [15]. Parameters concerning the spatial and temporal behavior of the applied current can be incorporated in the analysis in order to supplementary enhance the accuracy of the prediction. These parameters may include transient effects of the applied current, different shapes and areas of the current loops inside the box, etc.



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
