# Peer review of "Distance Scaling Method for Accurate Prediction of Slowly Varying Magnetic Fields in Satellite Missions"

_Geoscientific Instrumentation, Methods and Data Systems, 2015_

## Referee Comment (RC1) · Anonymous Referee #1 · 18 May 2016

General comment: The article demonstrates some peculiarities of measurements during magnetic cleanliness analyzing and interesting methods of estimation of magnetic interference generated by a set of sources localized in an unknown and random way. However, practical applicability of the suggested method is limited to the case when the point of interest is located between two magnetometers ("Extrapolation between the magnetometers") placed at big enough distance – in this case, one can get much better accuracy in comparison to the widely used model of equivalent dipole. Nevertheless, knowledge of the equivalent dipole parameters is important for prediction of attitude of a satellite (in low Earth orbit).

Major comments: Why values of magnetic field calculated using different approaches

(both theoretical, actually) are named as "predicted" and "theoretical" (p. 5, line 21) and the difference between them is used as criterion of "quality" of predictions? Especially taking into account next statement: "For low frequencies ($\omega \to 0$), the vector potential theory and the Biot – Savart law can be used interchangeably" (p.5, l. 4), as well as the fact that all modelling and simulations in the article have been done for DC magnetic field.

Concerning section B. "Extrapolation at larger distances using smoothing technique", it would be interesting to compare results shown in Fig. 5 with equivalent dipole filed - the $r^{-3}$ dependence.

Why Eq. 8 "signifies that the magnetic field fall-off will exhibit the $r^{-3}$ distance power law dependence at smaller distances" (p. 8, l.1)?

Other results are rather trivial – errors decrease when the ratio distance to the EUT's size increase.

Minor comments:

The abbreviation EUT should be explained when used the first time

The statement about magnetic sources as dipoles: "However, this assumption is not always valid since several parameters of the magnetic sources may have significant impact on the EUT's magnetic signature." (p. 1, l. 33) is probably excessive here since it is explained below on p. 2, l. 20.

---

## Referee Comment (RC2) · Anonymous Referee #2 · 19 May 2016

This study presents a novel method to accurately determine the magnetic behavior of an equipment under test (EUT) with slowly (up to 1 mHz) varying fields. The authors also validates their proposed method by applying it to real measurements. While this manuscript has the potential to be published in GI, my major concern is that the authors did not clearly state under what conditions each of their proposed method (exponential interpolation and the smoothing technique) is applicable, or more suitable. Currently in the manuscript, the authors just showed the prediction accuracy of each technique in various tests, but did not properly summarize the advantage of each proposed method. More clarifications on this regard would improve the manuscript. I suggest they include some discussions/comparisons before Section 3.F (maybe using a table to list the advantage of each technique with improvements from previous methods). Please find a list of my other minor comments below.

1. In the abstract, please include the full name before using the acronym (such as EUT and SNR).

2. In Line 2 of page 2, I believe the authors were trying to say "DC-1 mHz", instead of DC to megahertz (MHz)?

3. In Line 4 of page 2, I suggest "at the specification points" be changed to "at specific points".

---

## Author Comment (AC1) · 31 May 2016

We would like to thank the Reviewers for their comments and suggestions that helped us to clarify several important issues and significantly enhance the quality of our paper.

REVIEWER 1 We would like to thank Reviewer 1 for his/her comments and suggestions that helped us to clarify important critical issues. Based on these comments/suggestions, several changes to the paper have been made.

1. General comment: The article demonstrates some peculiarities of measurements during magnetic cleanliness analyzing and interesting methods of estimation of magnetic interference generated by a set of sources localized in an unknown and random

way. However, practical applicability of the suggested method is limited to the case when the point of interest is located between two magnetometers ("Extrapolation between the magnetometers") placed at big enough distance – in this case, one can get much better accuracy in comparison to the widely used model of equivalent dipole. Nevertheless, knowledge of the equivalent dipole parameters is important for prediction of attitude of a satellite (in low Earth orbit).

1a. As the reviewer suggests, in order to demonstrate the practical applicability of the suggested methods several revisions are proposed and described under his/hers comments. The article focuses on the modeling of the magnetic field of an Equipment Under Test (EUT) based on near magnetic field measurements (on-ground). These measurements are typically performed in the near – field area of the EUT for high SNR purposes. Then, a model is created that is destined to emulate the magnetic behavior of the EUT in the spacecraft. A dominant technique involves the Magnetic Dipole Modeling (MDM), where the magnetic sources are traditionally treated as dipoles and their parameters (position, magnetic moment) are estimated [1, 4, 7]. However, in cases where the EUT consists of several magnetic sources, the equivalent MDM technique fails to provide a reliable model which can be later used for extrapolation purposes [4, 15]. Moreover, the number of magnetometers needed for this type of modeling is quite large. Therefore, a two magnetometer technique is widely employed [14] in order to estimate the fall off of the magnetic field of an EUT.

1.1. Major comments: Why values of magnetic field calculated using different approaches (both theoretical, actually) are named as "predicted" and "theoretical" (p. 5, line 21) and the difference between them is used as criterion of "quality" of predictions? Especially taking into account next statement: "For low frequencies ($k \rightarrow 0$), the vector potential theory and the Biot – Savart law can be used interchangeably" (p.5, l. 4), as well as the fact that all modelling and simulations in the article have been done for DC magnetic field.

1.1a. As the reviewer suggests, the terms "predicted" and "theoretical" magnetic field

have been renamed to B_estimated and B_actual respectively. B_actual is calculated via Eq.2 or Eq.4 and represents the real value of the magnetic field at specific points. B_estimated represents the modeled magnetic field which is calculated (extrapolated) using Eq. 5. The magnetic field values at two observation points (magnetometers) are adequate to determine the coefficients a and n. Thus, Eq. 5 can be used to estimate the magnetic field values at larger distances (B_estimated) from near field measurements (B_actual). Observation points are the location of the magnetometers (2 in the present work) that are used in order to measure the magnetic field of an EUT, as shown in Fig. 2. Specification points are the location of the measuring equipment in spacecraft coordinate system, typically mounted on a long boom, where magnetic cleanliness is required. The predicted values of the magnetic field B_estimatedare calculated at the specification points. Then, B_estimatedis compared to B_actualin order to assess the modeled power law of the magnetic field (Eq. 5). From a theoretical point of view, for low frequencies (k → 0), the vector potential theory and the Biot – Savart law can be used interchangeably, thus Eq. 2 and Eq. 4 will result in the same B_actual. In this work, virtual DC magnetic sources were simulated and their magnetic field was calculated employing the Biot – Savart law (Eq. 2). However, the magnetic field was also calculated in some cases employing the vector potential theory (Eq. 4). The results were cross – referenced and clearly show that the two above equations can be used interchangeably for purposes of magnetic field calculation at the lower frequency regime.

1.1b. In order to clarify the difference between "predicted" and "theoretical" magnetic field, the terms B_estimated and B_actual will be used throughout the paper. Moreover, in p. 5, l. 6 the sentence "For low frequencies (k → 0), the vector potential theory and the Biot – Savart law can be used interchangeably." will be replaced by "For low frequencies (k → 0), the vector potential theory and the Biot – Savart law can be used interchangeably and thus the magnetic field can be calculated using either Eq. 2 or Eq. 4 [9]." Finally, the following paragraph will be added (p. 5, l. 20) "B_actual is calculated via Eq.2 or Eq.4 and represents the real value of the magnetic field. B_estimated

represents the modeled magnetic field which is calculated (extrapolated) using Eq. 5. The magnetic field values at two observation points are adequate to determine the coefficients a and n. Observation points are the location of the magnetometers (2 in the present work) that are used to measure the magnetic field of an EUT, as shown in Fig. 2. Specification points are the location of the measuring equipment in spacecraft coordinate system, typically mounted on a long boom, where extrapolation of the magnetic field is required. Thus, Eq. 5 can be used to estimate the magnetic field at larger distances (B_estimated) from near field measurements (B_actual). Then, B_estimated is compared to B_actual at the specification points in order to assess the modeled power law of the magnetic field (Eq. 5). Specifically, the relative error (R_E) between estimated and actual field values is examined and is used as a criterion to assess the overall prediction process: R_E=(|B_estimated-B_actual | )/(|B_actual |)·100% (7) ”

1.2. Concerning section B. “Extrapolation at larger distances using smoothing technique”, it would be interesting to compare results shown in Fig. 5 with equivalent dipole field - the r^−3 dependence.

1.2a. The reviewer's comment is quite useful and interesting. Both figures 4 and 5 have been updated to compare results with equivalent dipole field - the r^−3 dependence.

1.2b. Figures 4 and 5 have been updated as shown below:

Fig.4. Magnetic field's average relative error with respect to the first magnetometer's position with fixed spacing 30cm, 40cm, 50cm and 60cm between the magnetometers. The dipole's field r^(-3) power law is also shown for comparison.

Fig.5. Magnetic field's average relative error with respect to the first magnetometer's position with fixed spacing 30cm, 40cm, 50cm and 60cm between the magnetometers, using the smoothing technique. The dipole's field r^(-3) power law is also shown for comparison.

1.3. Why Eq. 8 "signifies that the magnetic field fall-off will exhibit the r^−3 distance power law dependence at smaller distances" (p. 8, l.1)?

1.3a. n varies between 2.2 − 3.8 and converges to 3 as the observation distance increases. Therefore, by smoothing the magnetic field power law with (n+3)/2 in the denominator of Eq. 8, it is expected that the fall – off will converge to 3 at smaller observation distances, thus enabling the placing of the magnetometers even closer to the EUT.

1.3b. The sentence on p. 8, l.1 "The above equation signifies that the magnetic field fall-off will exhibit the r^(-3) distance power law dependence at smaller distances, thus enabling the placing of the magnetometers even closer to the EUT." is replaced with "The power law dependence n varies between 2.2 − 3.8 for small observation distances amongst the test scenarios examined, and converges to 3 as the observation distance increases. Therefore, by smoothing the magnetic field power law with (n+3)/2 in the denominator of Eq. 8, it is expected that the fall – off will converge to 3 at smaller observation distances, thus enabling the placing of the magnetometers even closer to the EUT."

1.4. Other results are rather trivial – errors decrease when the ratio distance to the EUT's size increase.

1.4a. The reviewer is correct. Errors decrease when the ratio observation distance to the EUT's size increases. However, in most cases magnetic field measurements need to be performed as close to the EUT as possible, for high Signal to Noise Ratio (SNR) purposes, even for EUTs with large dimensions.

1.4b. The aforementioned issue is already noted on p. 2, l. 18: "Typically, the magnetic field measurements are conducted in close proximity to the EUT for high Signal to Noise (SNR) purposes."

1.5. The abbreviation EUT should be explained when used the first time

1.5a. The reviewer is correct. It should be noted as "Equipment under Test".

1.5b. In the Abstract, on p. 1, l. 10 the EUT will be replaced by "Equipment under Test (EUT)"

1.6. The statement about magnetic sources as dipoles: "However, this assumption is not always valid since several parameters of the magnetic sources may have significant impact on the EUT's magnetic signature." (p. 1, l. 33) is probably excessive here since it is explained below on p. 2, l. 20.

1.6a. As the reviewer suggest, the statement is excessive since it is explained more explicitly on p. 2, l. 20.

1.6b. The sentence "However, this assumption is not always valid since several parameters of the magnetic sources may have significant impact on the EUT's magnetic signature." (p. 1, l. 33) will be removed.

REVIEWER 2

We would like to thank Reviewer 2 for his/her comments and suggestions that helped us to clarify important issues and enhance the quality of the paper. Based on these comments/suggestions, several changes to the paper are proposed.

2. This study presents a novel method to accurately determine the magnetic behavior of an equipment under test (EUT) with slowly (up to 1 mHz) varying fields. The authors also validate their proposed method by applying it to real measurements. While this manuscript has the potential to be published in GI, my major concern is that the authors did not clearly state under what conditions each of their proposed method (exponential interpolation and the smoothing technique) is applicable, or more suitable. Currently in the manuscript, the authors just showed the prediction accuracy of each technique in various tests, but did not properly summarize the advantage of each proposed method. More clarifications on this regard would improve the manuscript. I suggest they include some discussions/comparisons before Section 3.F (maybe using a table to list the advantage of each technique with improvements from previous methods). Please find a list of my other minor comments below.

2a. As the reviewer suggests, a new Section will be added before Section 3.F in order to illustrate the advantages of each proposed technique and highlight improvements from previous methods.

2b. A new Section before Section 3.F will be added as follows: F. Synopsis and Discussion The two methods that are proposed in the present work ensure better accuracy than previous methods in the estimation of the magnetic field at various distances, as readily observed from Figs. 4-5 and Figs. 7-9. Moreover, the location of the observation points (magnetometers) that intend to verify strict ground verification requirements of equipment mounted on a spacecraft is a critical issue, especially in cases where the magnetic signature of the EUT is comparable to the ambient noise. In particular, the magnetometers should be placed as close to the EUT as possible in order to achieve a high SNR. Specifically, the exponential interpolation technique enables the placing of the magnetometers close to the EUT, i.e. the first magnetometer can be located at approximately 2 times the maximum dimension of the box, regardless of the spacing between the two magnetometers. Similarly, the smoothing technique permits the placing of the magnetometers even closer, i.e. the first observation point can be placed at approximately 1 times the dimension of the box. It is worth mentioning that in all cases the relative error between estimated and actual field values is kept below 10%. Finally, it should be noted that both techniques were validated in scenarios when the box's maximum dimension is varied. As readily observed from Figs. 8-9, previous methods fail to provide an accurate prediction at larger distances resulting in over/underestimation of the magnetic field depending on the specific points of interest. Nevertheless, both proposed techniques result in adequate estimation of the magnetic field (less than 10% relative error between estimated and actual field values) regardless of the specification distance.

2.1. In the abstract, please include the full name before using the acronym (such as

EUT and SNR).

2.1a. The reviewer is correct. It should be noted as "Equipment under Test" and "Signal to Noise Ratio".

2.1b. In the Abstract, on p. 1, l. 10 EUT will be replaced by "Equipment under Test (EUT)" and p. 1, l. 13, SNR will be replaced by "Signal to Noise Ratio (SNR)".

2.2. In Line 2 of page 2, I believe the authors were trying to say "DC-1 mHz", instead of DC to megahertz (MHz)?

2.2a. Following the reviewer's comment, this issue needs further clarification. On p. 2, l. 2, the requirement for magnetic cleanliness at the frequency range "DC-1 MHz" is valid for recent and upcoming space missions, such as Solar Orbiter and Lisa Pathfinder [10, 11]. However, in the present work, magnetic sources at the lower frequency regime (up to 1 mHz) were employed in several cases to compose an EUT.

2.2b. On p. 1, l. 34, the sentence "Furthermore, upcoming space missions have additional demanding requirements regarding AC magnetic cleanliness, focusing on the range DC – 1 MHz, where the magnetic field can be treated as quasi-static [12]." will be replaced with "Furthermore, upcoming space missions, such as Solar Orbiter [10] and Lisa Pathfinder [11], have additional demanding requirements regarding AC magnetic cleanliness, focusing on the range DC – 1 MHz, where the magnetic field can be treated as quasi-static [12]." Moreover, in line 8 of page 2 the sentence "Low frequency (up to 1 MHz) magnetic sources have been employed in several cases to compose an EUT." will be replaced to "Low frequency (up to 1 mHz) magnetic sources have been employed in several cases to compose an EUT."

2.3. In Line 4 of page 2, I suggest "at the specification points" be changed to "at specific points".

2.3a. As the reviewer suggest, the phrase "at the specification points" will be changed to "at specific points".

GID

2.3b. On p. 2, l.4, the phrase "at the specification points" will be updated to "at specific points".

Please also note the supplement to this comment:
http://www.geosci-instrum-method-data-syst-discuss.net/gi-2015-47/gi-2015-47-AC1-supplement.pdf

———————————————————
[Figure]

**Fig. 1.** Fig.4. Magnetic field's average relative error with respect to the first magnetometer's position with fixed spacing 30cm, 40cm, 50cm and 60cm between the magnetometers. The dipole's field rˆ(-3) power

**Fig. 2.** Fig.5. Magnetic field's average relative error with respect to the first magnetometer's position with fixed spacing 30cm, 40cm, 50cm and 60cm between the magnetometers, using the smoothing technique.